# What Works in Community-Led Suicide Prevention: Perspectives of Wesley LifeForce Network Coordinators

**DOI:** 10.3390/ijerph18116084

**Published:** 2021-06-04

**Authors:** Lennart Reifels, Amy Morgan, Lay San Too, Marisa Schlichthorst, Michelle Williamson, Helen Jordan

**Affiliations:** 1Centre for Mental Health, Melbourne School of Population and Global Health, The University of Melbourne, Carlton, VIC 3010, Australia; ajmorgan@unimelb.edu.au (A.M.); tiffany.too@unimelb.edu.au (L.S.T.); schlichthorst@gmail.com (M.S.); mkwill@unimelb.edu.au (M.W.); 2Centre for Health Policy, Melbourne School of Population and Global Health, The University of Melbourne, Carlton, VIC 3010, Australia; h.jordan@unimelb.edu.au

**Keywords:** suicide, suicide prevention, community coalitions, community networks

## Abstract

Community coalitions have been recognised as an important vehicle to advance health promotion and address relevant local health issues in communities, yet little is known about their effectiveness in the field of suicide prevention. The Wesley Lifeforce Suicide Prevention Networks program consists of a national cohort of local community-led suicide prevention networks. This study drew on a nationally representative survey and the perspectives of coordinators of these networks to identify the key factors underpinning positive perceived network member and community outcomes. Survey data were analysed through descriptive statistics and linear regression analyses. Networks typically reported better outcomes for network members and communities if they had been in existence for longer, had a focus on the general community, and had conducted more network meetings and internal processes, as well as specific community-focused activities. Study findings strengthen the evidence base for effective network operations and lend further support to the merit of community coalitions in the field of suicide prevention, with implications for similar initiatives, policymakers, and wider sector stakeholders seeking to address suicide prevention issues at a local community level.

## 1. Introduction

Suicide is a major public and mental health concern in communities around the world [1,2], claiming the lives of 3318 Australians in 2019 [3]. It is estimated that for every life lost to suicide up to 80 relatives, friends and acquaintances can be affected [4], with the potential for profound impacts on the mental health and well-being of family members and those in the wider community and social networks [5,6].

Community-based suicide prevention approaches have been recognised as an important component of national suicide prevention strategies [7], yet little is known about their effectiveness [8,9]. The diversity of local community contexts and varying nature of suicidality issues presenting among affected groups highlight the importance of a locally tailored and well-coordinated approach to suicide prevention that works from the ground up to optimally address community needs [8]. Community coalitions can therefore provide a central mechanism to mobilise and coordinate relevant local stakeholder efforts and establish a whole-of-community approach to address local health issues [10,11], which has also gained increasing attention in the field of suicide prevention [12].

Notwithstanding broader insights from the field of health promotion, there is a dearth of evidence on the effectiveness of community coalitions in the field of suicide prevention. The few existing studies of community-led suicide prevention coalitions mainly examined implementation and capacity-building aspects, while an assessment of resulting outcomes and key factors underpinning effective coalition functioning is still largely missing today [12]. In this context, prior quantitative studies have demonstrated the effectiveness of suicide prevention training and capacity-building initiatives [13,14], while scarce assessments of broader coalition outcomes and effectiveness have often been reliant on qualitative data. This underscores the need to further strengthen the evidence base for effective community-led strategies in suicide prevention.

Challenges to establishing successful community coalitions with a view to promoting public health have been documented in the literature. In this context, Zakocs and Edwards noted that ‘Initiating and sustaining coalitions is no simple task… It is a complex, dynamic process that involves multiple coalition-building tasks, such as recruiting members, identifying lead agencies, generating resources, establishing decision-making procedures, fostering leadership, building the capacity of members to participate, encouraging consensus-based planning for action, implementing agreed-upon actions by negotiating with key stakeholders in the community, refining strategy based on evaluation data, and establishing mechanisms for institutionalizing coalitions and/or their strategies’ [10]. Such insights highlight the need to understand the key characteristics, internal processes, and community-focused activities of effective community coalitions in the field of suicide prevention.

Among the key factors found to underpin the effective functioning and internal and external outcomes of community and health coalitions are formal characteristics of coalition structure and processes of internal organisation [10,15]. Reviewing the literature on health networks, Cunningham et al. concluded that ‘more work is needed to demonstrate effectiveness, and to identify factors related to improved outcomes’ [16]. Beyond a prior research emphasis on formal network characteristics underpinning network effectiveness, it is therefore essential to further examine those community-focused suicide prevention activities which are associated with improved community outcomes [9].

### Wesley LifeForce Suicide Prevention Networks

Wesley Mission, through its Wesley LifeForce Suicide Prevention Networks program (Wesley LifeForce program), has been working with Australian communities to support the development of local community suicide prevention networks since 2007. Community networks are a common type of community coalition, and the term is often used synonymously with the latter in Australia. Wesley Mission defines a network as ‘a union of people and organisations, working together to change the outcome relating to a specific problem.’ Networks are further described as being community-led and as being ‘for the people, by the people’ [17]. While the aims and objectives of each network vary, reflecting the unique identity of each community, there is a common thread: a focus on interagency cooperation and raising community awareness. Their portfolio of activities is uniquely tailored to local community contexts and commonly aimed at upstream initiatives that are focused on awareness raising, stigma reduction, fostering help seeking, training, capacity building, and community development. The Wesley LifeForce program enables local network establishment through seed funding, facilitating initial community stakeholder consultations and network planning meetings, providing access to suicide prevention training, guidance on network governance and strategic planning, and through a dedicated team of community development staff who offer ongoing assistance to networks as needed. LifeForce networks typically consist of a core executive team and a broader membership comprising interested community members and key stakeholders from local organisations, services, and community groups with a mandate or interest in suicide prevention. Throughout the establishment process, LifeForce networks transition from initial community proposal and planning stages to becoming fully operational during incubating and sustainable stages. Over time, most LifeForce networks become independent, self-actuating community entities, which develop and implement locally targeted activities, projects, and services for suicide prevention, crisis intervention, and postvention [17]. Rather than providing a fixed or predetermined program of interventions that would uniformly apply across the board, networks provide a structural mechanism to develop and implement locally relevant solutions to advance suicide prevention in local communities.

The national scale of the Wesley LifeForce program, which comprises more than 104 local community grassroots networks across Australia, provides a unique opportunity to investigate the profile and key characteristics of effective community-led suicide prevention networks. This study, which formed part of a larger research project, was designed to address the evidence gap on effective community-led suicide prevention coalitions by addressing two key questions:What is the national profile of Wesley LifeForce Suicide Prevention Networks (in terms of network characteristics, internal processes, and community-focused activities)?Which network characteristics, internal processes, and community-focused activities are associated with better outcomes for network members and the community?

## 2. Materials and Methods

### 2.1. Study Design and Setting

The parameters of interest were initially informed by prior research literature on community coalitions and a program logic for LifeForce networks, outlining key internal network processes and anticipated or ‘expected’ member outcomes, community-focused network activities, and related community outcomes. To develop a nationally representative and current profile of key network characteristics, internal processes, and community-focused activities, and examine factors associated with reported network outcomes, a cross-sectional correlational study was conducted, involving an online survey of all operational Wesley LifeForce networks in Australia. The survey was targeted at the coordinating contact person within each network with the greatest knowledge of the network’s history and ongoing operations who would complete the survey on behalf of the network.

### 2.2. Survey Tool

The customised survey included a mix of closed and open-ended questions to capture information on respondent demographics, network characteristics, internal network processes, community-focused activities, as well as perceived outcomes for network members and the community. Respondents were asked whether their network had conducted any of a series of nine internal network processes or nine community-focused activities. Next, they were asked whether they had observed any changes resulting from these network processes and activities for network members and the community. The two main internal network outcomes examined in this study included perceived changes in network member understanding of key suicide prevention issues as well as member confidence and capacity to address such issues. Each of these network member outcomes was assessed through eight or seven survey items, respectively. Perceived external outcomes of network activities for communities, service providers, people at risk of suicide, and overall since the network began, were also examined, by way of four or five survey items, respectively. Respondents indicated on five-point Likert scales to what extent they agreed with observing changes in these outcome domains (with ratings being 1: strongly disagree, 2: disagree, 3: neutral, 4: agree, 5: strongly agree). A copy of the survey schedule is included in Appendix A.

### 2.3. Data Collection and Participant Recruitment

The online survey was administered via Qualtrics Research Core, an online survey engine, between September and October 2019. An email invitation to complete the survey was circulated by Wesley Mission on behalf of the research team to the primary contacts of all 92 sustainable and incubating networks among the 104 total LifeForce networks in existence at the time. Since the survey targeted only operational networks, networks at proposal or planning stages were not included. The initial invitation was followed by up to three fortnightly email reminders to non-responding networks. Survey completion took approximately 30–40 min.

Of the 92 invited network contacts, 47 responded to the survey (response rate 51.1%). Six responses were discarded due to incomplete data, leaving 41 responses to be included in data analysis (response rate 44.6%). The profiles of invited, responding, and non-responding networks showed a substantial level of concordance (Appendix A), which was indicative of the sample’s national representativeness.

### 2.4. Data Analysis

Quantitative data from closed survey questions and rating scales were analysed through descriptive statistics (frequencies, percentages, mean, median) to generate an overview of key network characteristics and activities as well as perceived outcomes for network members and the community. Univariate linear regression analyses were conducted to examine key factors underpinning perceived network member and community outcomes. Predictor variables examined for network member outcomes included network characteristics and internal network processes, while predictor variables for community outcomes included network characteristics and community-focused network activities. Composite outcome variables, which reflected the mean score of constituent items, were generated for each main network internal outcome dimension (member understanding, member confidence/capacity) and external outcome domain (community, service providers, people at risk of suicide, overall). The composite outcomes scales utilised in this study showed good to excellent internal consistency, as evidenced by Cronbach’s alpha scores ranging from 0.84 to 0.94. Inferential data analyses were adjusted for multiple comparisons, using the Holm method [18]. All data analyses were conducted using Stata 16 (StataCorp, College Station, TX, USA).

### 2.5. Ethics

All survey respondents provided informed consent to participate in the study via the Qualtrics online platform. The study received ethical approval from the Human Research Ethics Committee at the University of Melbourne (ID: 1954813.3).

## 3. Results

### 3.1. Network and Respondent Profile

Table 1 outlines the key characteristics of the resulting network sample and survey respondents. Most networks were located in regional areas, followed by major cities and remote areas. All (but four) networks were classified as being sustainable. More than three-quarters of networks had a primary work focus on the general community, while five networks specifically focused on Indigenous communities, and one on culturally and linguistically diverse communities.

Additional data indicated that 17 networks had obtained additional funding support beyond seed funding to sustain their ongoing network operations. Networks had met between 2 and 16 times in the past 12 months, with an average of 9.3 meetings per year. Network membership numbers varied between 2 and 2600 (Median 14, IQR 10–20). The large variation can be explained by some respondents only counting active core members, while others considered the network membership more broadly. The number of local organisations, services, and community groups represented in the networks varied between 0 and 60 (Median 8, IQR 4–12). Approximately, 58.7% of all network members were reportedly involved as volunteers, while 40.0% participated as part of their professional role or employment. All, but one, of the 41 networks had members with a lived experience of suicide, with an average of 58.3% of network members estimated to share this experience.

The profile of survey respondents indicated that most were involved in network executive committees, while others were general network members. Just over one-half were involved on a volunteer basis, with others involved as part of their professional role or employment. Network membership tenure varied between 1 and 14 years, with 80.7% involved between 1 and 5 years.

Additional data indicated that most survey respondents (85.4%) identified as having a lived experience of suicide. The majority had initially joined the network because they saw a need in the community and wanted to help create change (65.9%), while other motivations for joining the network were driven by the person’s lived experienced of suicide (14.6%) or their professional interest (17.1%). The average weekly hours contributed by respondents to the networks varied between 1 and 40 (mean 8.3).

### 3.2. Internal Network Processes and Community-Focused Activities

Table 2 outlines the internal network processes and community-focused activities conducted by LifeForce networks over the past 12 months and/or more than 12 months ago. Overall, more than two-thirds of participating networks indicated that they had engaged in each of the nine listed internal processes over the past year. Among these, the processes most frequently conducted included identifying relevant community stakeholders (87.8%), identifying suicide prevention issues in the community (85.4%), identifying gaps in community knowledge on suicide (78.0%), and training and capacity building of network members (75.6%). The internal processes most frequently conducted more than 12 months ago involved identifying local service needs, gaps, and access barriers (31.7%) and identifying local service arrangements and referral pathways (29.3%).

Overall, networks had been involved in an average of 7.2 (SD 2.4) of the nine listed community-focused activities. Among these, the activities most frequently conducted over the past year were distributing support service contact information (85.4%), facilitating community suicide awareness and stigma reduction initiatives (85.4%), as well as initiating suicide prevention activities that address the needs of diverse populations and service gaps (78.1%) and facilitating access to suicide prevention training (78.1%). Activities conducted to a lesser extent (by just over one-half of networks) were those involving advocating and promoting responsive service policies, proactive guidelines, and appropriate referral pathways (51.2%) and facilitating bereavement support and postvention activities (51.2%).

### 3.3. Perceived Network Member Outcomes

Table 3 outlines the mean score of perceived outcomes of internal processes for network members. Network coordinators consistently ‘agreed’ with observing positive network member outcomes, both in terms of improved member understanding of suicide prevention issues and their confidence and capacity to address these issues via targeted strategies. Notwithstanding relatively little variation in mean item ratings, perceived gains in network member understanding were most pronounced in relation to existing gaps in community knowledge regarding suicide and key suicide prevention issues in the community. Perceived gains in member confidence and capacity were most pronounced in the planning of initiatives that encouraged help seeking and initiatives to strengthen community responses. In addition to perceived network member outcomes, most survey respondents (87.8%) also indicated that being part of the network had made a positive difference to their own life.

### 3.4. Perceived Community Outcomes

Table 4 outlines the mean score of perceived outcomes of network activities for communities, for service providers, for people at risk of suicide, and overall since the network began. Overall, network coordinators consistently ‘agreed’ with observing positive network outcomes for communities and service providers, while perceived outcomes for people at risk of suicide and overall network outcomes were more nuanced. Survey respondents consistently ‘agreed’ with seeing positive outcomes of network activities for communities in terms of increased knowledge about support services and their linkages, improved awareness of suicide prevention services/strategies, and increased confidence and capacity to help people at risk of suicide. Similarly, respondents consistently ‘agreed’ with observing positive outcomes for service providers, which was evident in improved awareness of suicide prevention services/strategies, increased confidence and capacity to help people at risk of suicide, greater inclusion of people with a lived experience in suicide prevention activities, and improved service linkages and access pathways.

Respondent ratings of perceived outcomes for people at risk of suicide were overall slightly more cautious, with mean scores in the ‘neutral’ to ‘agree’ range. The aspects rated closest to ‘neutral’ included the early identification of people at risk of suicide and appropriate service support and referral. Outcome aspects with a slightly greater tendency towards agreement included less experience of stigma and better support in the community. By and large, respondent ratings in this outcome domain indicated that outcomes for people at risk of suicide were perceived to be comparatively unchanged. Similarly, when asked about overall outcomes observed since the network began, respondents tended to indicate no significant status changes in terms of a reduction of suicide risk in the community or improved community well-being. Nonetheless, other outcome aspects in the agreement range included the improved coordination of suicide prevention efforts, which also followed more of a whole-of-community approach. While the results above are based on mean scores for all participating networks, important differences in the outcomes observed between networks are further explored in the following.

### 3.5. Predictors of Member Outcomes

Table 5 shows the results from the univariate linear regression analysis on the network characteristics and internal network processes that predicted better-perceived outcomes for network members, with findings that remained statistically significant after adjustment for multiple comparisons denoted with an asterisk.

Key network characteristics and internal processes linked to improvements in network member understanding of suicide prevention issues included the greater frequency of network meetings and a greater number of internal network processes conducted. In turn, greater network meeting frequency, a primary focus on the general community, and earlier stakeholder identification were also linked to perceived increases in member confidence and capacity to address suicide prevention issues.

All factors identified as significant above were also examined in multiple regressions. However, due to the smaller resulting sample size (*n* = 28) of networks with consistent data across all variables, these analyses were underpowered. Findings, therefore, need to be interpreted with caution. After controlling for other significant factors, the only network characteristic that showed a significant positive relationship with member outcomes (in terms of greater confidence and capacity) was a network focus on the general community (rather than a specific target group).

### 3.6. Predictors of Community Outcomes

Table 6 shows the results from the univariate linear regression analysis on the network characteristics and community-focused activities that predicted positive perceived outcomes for communities, service providers, people at risk of suicide, and overall. Findings that remained statistically significant after adjustment for multiple comparisons are denoted with an asterisk.

Network characteristics that were associated with positively perceived community outcomes included earlier years of network seed funding and greater meeting frequency. Several community-focused network activities were linked to positively perceived community outcomes. Networks that were engaged in facilitating community suicide awareness and stigma reduction initiatives or facilitating community access to support services were more likely to report positive community outcomes. Similarly, positively perceived community outcomes were more likely to be reported by networks which had supported services in their capacity to identify, respond to and assist suicidal people; had advocated and promoted service policies, guidelines, and referral pathways; and had conducted a greater number of network activities. The only network activity associated with positively perceived outcomes for service providers was facilitating community access to support services, while advocating and promoting service policies, guidelines, and referral pathways was linked to better perceived overall outcomes.

When significant predictor variables identified above were entered into multiple regression models for each outcome domain, the only variable that remained significant was the year of network seed funding with better community outcomes reported for older networks. None of the community-focused activities remained predictive of external network outcomes. This may again partly be due to the analysis being underpowered in view of the small sample size.

## 4. Discussion

This study drew on a nationally representative sample of LifeForce networks and the rich knowledge of network coordinators to advance the evidence base on effective practices in community-led suicide prevention. Study findings provide the first national profile of LifeForce networks and shed light on key factors underpinning perceived network member and community outcomes.

Most LifeForce networks had adopted multiple internal processes to organise the network and build the capacity of its members [19]. Survey findings provided indications of positive perceived outcomes resulting from such processes for network members, both in terms of improved member understanding of suicide prevention issues and increased confidence and capacity to address such issues via targeted strategies. Simultaneously, networks had conducted a variety of community-focused activities to address suicide prevention issues in local communities [8]. Overall, findings provided indications of positively perceived outcomes resulting from such network activities for communities and service providers, while perceived outcomes for people at risk of suicide and overall since the network began were more nuanced.

Perceived community outcomes included increases in knowledge and awareness of support services, and confidence and capacity to help someone at risk of suicide. Perceived outcomes for service providers related to improved awareness of suicide prevention services/strategies, increased confidence and capacity to help people at risk of suicide, the inclusion of people with a lived experience in suicide prevention activities, and improved service linkages and access pathways. Outcomes for people at risk of suicide were seen by many to have remained largely unchanged (with no evident gains reported in the early identification or receipt of appropriate service support and referral). While the local coordination of suicide prevention efforts had reportedly improved and followed more of a whole of community approach, overall levels of community suicide risk and well-being were equally seen to have remained largely unchanged. It is possible that these latter findings may partly reflect the difficulty for participants to confidently assess or rate such aspects and highlight the multifaceted nature of such issues, which may require concerted approaches at multiple levels [20] and which may be hard to shift by means of a single suicide prevention initiative. There is merit in bolstering the evaluation capacity of the LifeForce program through a national approach to data collection and developing the evaluation expertise of networks at a community level [21].

Exploratory analyses shed further light on network characteristics and activities that were associated with positive perceived network outcomes [10]. Networks which had been in existence for longer, which had a focus on the general community, and which had conducted more meetings and internal processes typically tended to report better outcomes for network members and communities. Such characteristics are reflective of broad-based networks with a degree of internal organisation and effective functioning that has previously been linked to positive coalition outcomes [22]. Greater meeting frequency, earlier stakeholder identification, and a general community focus also underpinned greater member confidence and capacity to address local suicide prevention issues, as an indicator of programmatic capacity [19]. Network activities linked to positively perceived community outcomes involved common suicide prevention strategies such as facilitating suicide awareness and stigma reduction initiatives [8] and community access to support services. The latter activity was also linked to perceived benefits for service providers, while advocating and promoting service policies, guidelines, and referral pathways was linked to better perceived overall outcomes. Notably, several predictors of positive community outcomes revolved around the network’s role in fostering service access, referral pathways, and response capacity, which can serve a critical function in overcoming barriers to help seeking and service access [23].

In interpreting the study findings, it is important to note that these reflect the perspectives of network coordinators who are directly involved in running the network operations and therefore do not provide a direct account of community views or objective data on resulting network impacts. Nevertheless, factors linked to positively perceived network impacts across internal and external outcome domains were both plausible and resonated with the wider literature on community coalition effectiveness [10]. A systematic literature review on community coalitions in health promotion identified that those internal processes (or coalition-building factors) most consistently linked to greater coalition effectiveness included formalisation of rules and procedures, leadership style, member participation, membership diversity, agency collaboration, and group cohesion [10]. LifeForce networks that reach a degree of maturation in internal organisational processes, which maintain active linkages to key stakeholders, and access relevant sources of funding and support are arguably best placed to achieve positive outcomes in a sustainable way.

One of the unique features of LifeForce networks is that these form part of a vast national network of local suicide prevention initiatives, which are structurally supported through an overarching national program. Rather than being left to their own devices, local communities with identified suicide prevention needs are thereby actively supported by the LifeForce program through the process of network establishment. Initial network establishment processes that have been identified as critical in the wider literature, such as effective community stakeholder engagement [24], internal capacity building [19], and strategic planning [22], are thus directly supported and able to benefit from significant program level expertise. Notwithstanding such procedural support, the specific focus and nature of local suicide prevention activities are very much determined by the networks themselves, as are the future network directions. The program logic of the LifeForce networks informed, and was subsequently informed by, this study. Therefore, this research adds to the developing theory of change on suicide prevention networks.

The strong representation of people with lived experience of suicide in the network membership and among network coordinators reflects another unique characteristic of community coalition in the suicide prevention field. The value of lived experience expertise has been increasingly recognised as pivotal to the design and delivery of suicide prevention programs [25,26], and it is evident that LifeForce networks provide an effective means for engaging community members with a lived experience in suicide prevention. Notwithstanding some of these unique features, LifeForce networks also shared many characteristics of community coalitions in the broader health promotion field.

The fact that some key characteristics of successful networks were reflective of more established networks concords with a broader literature review on community suicide prevention interventions, which concluded that ‘only long-term programs that utilize a commitment of the society at multiple levels and succeed in establishing a community support network can effectively reduce suicidal rates’ [9]. This assessment underscores the importance of whole-of-community and whole-of-society approaches to suicide prevention. It also highlights the need to better understand key factors driving ongoing network sustainability beyond establishment as a key question for future research [27,28].

### Study Strengths and Limitations

The profile of participating networks was representative of the national cohort of LifeForce networks, which reflects a study strength and instils confidence in the findings. The cross-sectional study design and resulting sample size enabled exploratory analyses of key variable associations with perceived network internal and external outcomes but did not permit causal inferences or multiple regression analyses to control for other factors. Moreover, perceived network outcomes were assessed retrospectively and did not involve before and after assessments or control group comparisons. The survey data are directly informed by the perspectives of network coordinators who are intimately involved in running the network operations and therefore do not provide a direct account of the views of or impacts on community members, service providers, or those at risk of suicide. Future research may therefore benefit from soliciting the perspectives of multiple respondents within each network, and from triangulating survey findings with other data sources on network internal and community outcomes [11].

## 5. Conclusions

These study findings strengthen the evidence base for effective network operations and lend further support to the merit of community coalitions in the field of suicide prevention, with implications for similar initiatives, policymakers, and wider sector stakeholders seeking to address suicide prevention issues at a local community level. Specifically, the findings are instructive as these point to the types of structural coalition characteristics, internal processes, and community-focused activities that are associated with positive perceived member and community outcomes.

## Figures and Tables

**Table 1 ijerph-18-06084-t001:** Characteristics of participating networks and respondents (*N* = 41).

Networks	*n*	%	Respondents	*n*	%
**Status**			**Network role**		
Incubating	4	9.76	Chair/President	19	46.34
Sustainable	37	90.24	Secretary	9	21.95
**Focus**			Member	6	14.63
General	34	82.93	Project Officer	2	4.88
Indigenous	5	12.20	Treasurer	1	2.44
CALD	1	2.44	Other	4	9.76
Unknown	1	2.44	**Type of involvement**		
**Location**			Volunteer basis	22	53.66
New South Wales	12	29.27	Professional role	9	21.95
Queensland	9	21.95	Both (prof & volunteer)	10	24.39
Victoria	9	21.95	**Years with network**		
Western Australia	4	9.76	1 to 2 years	18	43.90
Northern Territory	3	7.32	3 to 5 years	11	36.83
Tasmania	3	7.32	6 to 10 years	9	21.96
South Australia	1	2.44	≥11 years	3	7.32
**Rurality**			**Age**		
Regional area	18	43.90	20 to 29 years	4	9.76
Major city	13	31.71	30 to 39 years	5	12.20
Remote area	10	24.39	40 to 49 years	9	21.95
**Year of seed funding**			50 to 59 years	13	31.71
2007–2011	5	12.2	60 to 69 years	7	17.07
2012–2016	11	26.8	70 to 79 years	3	7.32
2017–2019	15	36.6	**Gender identity**		
Yet, to be funded	1	2.4	Female	26	63.41
Unknown	9	22.0	Male	15	36.59
			**Indigeneity**		
			No	36	87.80
			Yes, Aboriginal	5	12.20

**Table 2 ijerph-18-06084-t002:** Internal network processes and community-focused activities conducted by networks (*N* = 41).

	In the Past 12 Months		More than 12 Months Ago		Never/Not Sure	
Internal Network Processes	*n*	%	*n*	%	*n*	%
Identifying suicide prevention issues in the community	35	85.4	7	17.1	1	2.4
Identifying available suicide prevention frameworks	30	73.2	10	24.4	3	7.3
Identifying local services and referral pathways	30	73.2	12	29.3	1	2.4
Identifying relevant community stakeholders	36	87.8	6	14.6	0	0.0
Identifying gaps in community knowledge on suicide	32	78.0	9	22.0	1	2.4
Identifying local service needs, gaps, and access barriers	28	68.3	13	31.7	3	7.3
Identifying gaps in suicide prevention efforts	29	70.7	10	24.4	4	9.8
Training and capacity building of network members	31	75.6	7	17.1	5	12.2
Participating in strategic planning to determine future network directions and activities	28	68.3	8	19.5	6	14.6
**Community-Focused Activities**						
Distributing support service contact information	35	85.4	8	19.5	2	4.9
Facilitating community suicide awareness and stigma reduction initiatives	35	85.4	6	14.6	4	9.8
Fostering recognition and capacity of lived experience in suicide prevention	27	65.9	6	14.6	11	26.8
Initiating suicide prevention activities that address the needs of diverse populations and service gaps	32	78.1	6	14.6	7	17.1
Supporting services to build their capacity in identifying, responding to, and assisting suicidal people	25	61.0	8	19.5	11	26.8
Facilitating access to suicide prevention training	32	78.1	7	17.1	5	12.2
Facilitating community access to support services	31	75.6	7	17.1	6	14.6
Advocating and promoting responsive service policies, proactive guidelines, and appropriate referral pathways	21	51.2	9	22.0	13	31.7
Facilitating bereavement support and postvention activities	21	51.2	8	19.5	15	36.6

Note: Percentages for row items may not add to 100% as networks could have conducted an activity or process both in the past 12 months and/or more than 12 months ago (thus both time variables were rated independently and not mutually exclusive).

**Table 3 ijerph-18-06084-t003:** Perceived network member outcomes (understanding, confidence, and capacity) (*N* = 41).

	Mean ^a^	Standard Deviation
**Better Understanding of:**		
Key suicide prevention issues in the community	4.02	0.65
Existing suicide prevention frameworks	3.80	0.60
Local service arrangements and referral pathways	3.83	0.86
Gaps in community knowledge regarding suicide	4.07	0.69
Local service needs and gaps	3.95	0.84
Gaps in suicide prevention efforts	3.93	0.75
Help-seeking barriers and facilitators	3.93	0.61
Service access barriers and facilitators	3.90	0.70
Overall understanding scale ^b^	3.93	0.50
**Increased Confidence and Capacity to:**		
Collaboratively plan and develop network-initiated strategies	4.00	0.77
Plan initiatives to address knowledge gaps	3.98	0.72
Plan initiatives to address service gaps and access issues	3.80	0.78
Plan initiatives to encourage help seeking	4.15	0.69
Plan initiatives to strengthen community responses	4.12	0.68
Plan initiatives to enhance service responses	3.73	0.71
Evaluate network initiatives	3.73	0.81
Overall confidence and capacity scale ^c^	3.93	0.58

^a^—Higher scores indicate stronger agreement: 1: strongly disagree, 2: disagree, 3: neutral, 4: agree, 5: strongly agree; ^b^—Cronbach’s alpha = 0.85; ^c^—Cronbach’s alpha = 0.90.

**Table 4 ijerph-18-06084-t004:** Perceived community outcomes resulting from network activities (*N* = 41).

	Mean ^a^	Standard Deviation
**The Community Now Has:**		
Increased knowledge about support services and their linkages	3.98	0.85
Improved awareness of suicide prevention services/strategies	3.98	0.79
Increased confidence in assisting people at risk of suicide	3.95	0.86
Increased capacity to respond and help someone at risk of suicide	3.88	0.78
Overall community scale ^b^	3.95	0.73
**Service Providers Now Have:**		
Improved awareness of suicide prevention services/strategies	3.95	0.67
Increased confidence in assisting people at risk of suicide	3.95	0.74
Increased capacity to respond and help someone at risk of suicide	3.93	0.75
Included people with a lived experience in suicide prevention activities	3.93	0.85
Improved service linkages and access pathways	3.76	0.73
Overall service provider scale ^c^	3.90	0.67
**People at Risk of Suicide:**		
Experience less stigma in the community	3.49	0.81
Are identified early	3.22	0.69
Receive appropriate support and referral to relevant support services	3.39	0.86
Demonstrate increased help-seeking behaviour and uptake of services	3.41	0.67
Are better supported by people in the community	3.46	0.78
Overall people at risk of suicide scale ^d^	3.40	0.61
**Overall, since the Network Began:**		
Suicide risk in the community has been reduced	2.73	0.84
The coordination of suicide prevention efforts has improved	3.73	0.81
Suicide prevention efforts follow a whole-of-community approach	3.80	0.95
Community well-being has improved	3.15	0.82
Overall outcomes scale ^e^	3.35	0.70

^a^—Higher scores indicate stronger agreement: 1: strongly disagree, 2: disagree, 3: neutral, 4: agree, 5: strongly agree; ^b^—Cronbach’s alpha = 0.91; ^c^—Cronbach’s alpha = 0.94; ^d^—Cronbach’s alpha = 0.86; ^e^—Cronbach’s alpha = 0.84.

**Table 5 ijerph-18-06084-t005:** Predictors of perceived network member outcomes (univariate linear regressions).

	Network Member Understanding		Network Member Confidence/Capacity	
	*b*	*p*	*b*	*p*
**Network Characteristics**				
Year of funding ^a^	−0.06	0.013	−0.06	0.063
Network received additional funding	0.35	0.028	0.06	0.744
Number of members ^b^	0.02	0.026	0.02	0.030
Proportion of volunteers	0.04	0.843	−0.16	0.520
Proportion of professionals	−0.26	0.218	0.11	0.655
Proportion of lived experience ^c^	−0.35	0.244	−0.50	0.144
Primary focus (specific group vs. general population)	−0.22	0.303	−0.65	<0.001 *****
Number of meetings in past year	0.07	0.006 *****	0.08	0.007 *****
**Internal Network Processes ^d^**				
Identifying suicide prevention issues in the community	0.70	0.173	0.66	0.269
Identifying available suicide prevention frameworks	0.60	0.045	0.03	0.938
Identifying local services and referral pathways	0.70	0.173	0.66	0.269
Identifying relevant community stakeholders ^e^	−0.65	0.005	−0.80	0.003 *****
Identifying gaps in community knowledge on suicide	0.70	0.173	0.66	0.269
Identifying local service needs, gaps, and access barriers	0.37	0.218	0.34	0.344
Identifying gaps in suicide prevention efforts	0.41	0.125	0.08	0.796
Training and capacity building of network members	0.01	0.982	0.34	0.222
Participating in strategic planning	0.45	0.039	0.53	0.038
Count of processes	0.21	0.005 *****	0.19	0.041

^a^—Data available for 31 networks; ^b^—excludes four outlier networks with number of members ≥ 80; ^c^—data available for 30 networks; ^d^—engaged in during the past 12 months or more than 12 months ago, versus ‘Never’ or ‘Not Sure’; ^e^—comparison is during the past 12 months versus more than 12 months ago, as no respondents selected ‘Never’ or ‘Not Sure’; *—estimate remained statistically significant after adjusting for multiple comparisons.

**Table 6 ijerph-18-06084-t006:** Predictors of perceived community outcomes (univariate linear regressions).

	Community Outcomes		Service Providers		People at Risk of Suicide		Overall Outcomes	
	*b*	*p*	*b*	*p*	*b*	*p*	*b*	*p*
**Network Characteristics**								
Year of funding ^a^	−0.13	0.000 *****	−0.07	0.012	−0.02	0.357	−0.07	0.011
Network received additional funding	−0.08	0.728	−0.13	0.532	−0.01	0.948	−0.35	0.114
Number of members ^b^	0.02	0.160	0.03	0.015	0.03	0.015	0.00	0.768
Proportion of volunteers	−0.09	0.761	−0.09	0.761	0.03	0.921	−0.41	0.156
Proportion of professionals	0.19	0.542	0.16	0.590	0.05	0.853	0.27	0.376
Proportion of lived experience ^c^	−0.63	0.198	−0.85	0.074	−0.99	0.018	−0.65	0.147
Primary focus (specific group vs. general population)	−0.45	0.139	−0.47	0.035	−0.03	0.912	−0.43	0.046
Number of meetings in past year	0.11	0.003 *****	0.08	0.015	0.07	0.025	0.05	0.193
**Network Activities ^d^**								
Distributing support service contact information ^e^	0.99	0.059	0.63	0.196	0.42	0.355	0.9	0.078
Facilitating community suicide awareness and stigma reduction initiatives	1.05	0.005 *****	0.45	0.210	0.27	0.405	0.74	0.044
Fostering recognition and capacity of lived experience in suicide prevention	0.58	0.023	0.16	0.492	0.17	0.444	0.23	0.349
Initiating suicide prevention activities that address needs of diverse populations and service gaps	0.41	0.182	0.47	0.092	0.37	0.143	0.25	0.39
Supporting services to build their capacity in identifying, responding to and assisting suicidal people	0.67	0.007 *****	0.59	0.011	0.37	0.089	0.55	0.026
Facilitating access to suicide prevention training	0.51	0.148	0.44	0.176	0.36	0.223	0.46	0.173
Facilitating community access to support services	0.86	0.006 *****	0.82	0.004 *****	0.54	0.043	0.71	0.021
Advocating and promoting responsive service policies, proactive guidelines and appropriate referral pathways	0.76	0.001 *****	0.58	0.008	0.29	0.166	0.71	0.002 *****
Facilitating bereavement support and postvention activities	0.31	0.197	0.25	0.263	−0.01	0.97	0.22	0.349
Count of activities	0.15	0.001 *****	0.11	0.011	0.07	0.105	0.11	0.012

^a^—Data available for 31 networks; ^b^—excludes four outlier networks with number of members ≥ 80; ^c^—data available for 30 networks; ^d^—engaged in during the past 12 months or more than 12 months ago, versus ‘Never’ or ‘Not Sure’; ^e^—as most networks (95%) had conducted this activity, insufficient network variation may have precluded this factor from becoming significant; *—estimate remained statistically significant after adjusting for multiple comparisons.

## Data Availability

The raw data used for this article are available upon reasonable request in writing to the corresponding author.

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
