# Peer review of "What Works in Community-Led Suicide Prevention: Perspectives of Wesley LifeForce Network Coordinators"

_ijerph, 2021, doi:10.3390/ijerph18116084_

Round 1
Reviewer 1 Report
Some suggestions for authors to consider:
- Why were network characteristics, internal network processes, network/community-focused activities treated as predictors of perceived community outcomes? The authors should explain the interrelationships among the variables.
- There seem some gaps in the perception of network process and community outcomes between the service providers and people at risk of suicide. What causes such gaps?
- Why was year of funding associated with negative community outcomes for network member understanding and service providers?
Author Response
Dear Editors and Reviewers,
We greatly appreciate the constructive feedback on our manuscript ‘What works in community-led suicide prevention: Perspectives of Wesley LifeForce network coordinators’.
For your consideration, please now find the revised manuscript attached, alongside a point-by-point response addressing the reviewer feedback below. All resulting changes to the revised manuscript have been highlighted in tracked changes. We believe that the revised manuscript has further gained in clarity and been strengthened considerably.
On behalf of the authors.
Yours sincerely,
Dr Lennart Reifels
Reviewer 1:
Some suggestions for authors to consider:
- Why were network characteristics, internal network processes, network/community-focused activities treated as predictors of perceived community outcomes? The authors should explain the interrelationships among the variables.
RESPONSE:
Thank you for these suggestions. Consistent with prior research on community coalitions in the broader health promotion field, a primary aim of our study was to identify those network key features (including, network characteristics, processes, and activities) that were associated with positive perceived network member and community outcomes. Further information on prior research in this area, including key predictor and outcomes variables, has now been included in Section 1. This being a cross-sectional study, it is also important to note that we are primarily looking at associations (and not causal relationships) between variables, which has been clarified throughout the manuscript and in Section 4.1.
- There seem some gaps in the perception of network process and community outcomes between the service providers and people at risk of suicide. What causes such gaps?
RESPONSE:
You are right. Perceived community outcomes resulting from network activities (outlined in Table 4) do indeed vary slightly for service providers and people at risk of suicide, with respective mean ratings ranging between 3.76-3.95 for the former and 3.22-3.49 for the latter. It is important to note that all of these ratings are based on respondent perceptions. While the reasons for these variations are not fully known, we speculate that these may be partly due to two reasons: a) difficulties for respondents to confidently assess or rate such aspects, and b) the complexity of real-world service delivery systems, which may be hard to change by means of single upstream initiative. To this end, the related Discussion section also states that:
“Outcomes for people at risk of suicide were seen by many to have remained largely un-changed (with no evident gains reported in the early identification or receipt of appropriate service support and referral). While the local coordination of suicide prevention efforts had reportedly improved and followed more of a whole of community approach, overall levels of community suicide risk and wellbeing were equally seen to have remained largely unchanged. It is possible that these latter findings may partly reflect the difficulty for participants to confidently assess or rate such aspects and highlight the multifaceted nature of such issues, which may require concerted approaches at multiple levels and which may be hard to shift by means of a single suicide prevention initiative. There is merit in bolstering the evaluation capacity of the LifeForce program through a national approach to data collection and developing the evaluation expertise of networks at a community-level.”
- Why was year of funding associated with negative community outcomes for network member understanding and service providers?
RESPONSE:
To clarify this slightly counter-intuitive point, the negative association between year of network seed funding and network member understanding indicates that members of networks with earlier seed funding (i.e., networks which had been in existence for longer) had a greater perceived understanding of suicide prevention issues. The nature and direction of this association would therefore seem entirely plausible. Similarly, it can be expected that more established networks (i.e., with earlier years of seed funding) have a better chance of demonstrating positive perceived community outcomes.
Reviewer 2 Report
Comments to the Author
This manuscript, titled “What works in community-led suicide prevention: Perspectives of Wesley LifeForce network coordinators,” examined the predictors of the effectiveness of the Wesley Lifeforce Suicide Prevention Networks program. The researchers recruited the coordinators of these networks who provided their perspectives on the factors that determined the outcomes of the program. The results identified several significant predictors, such as strategic planning and additional funding.
The study examined an important research question. However, there are several major issues that need to be addressed.
- The study should be theory-driven, and the literature review should discuss in relation to theories and past literature. The study had clear aims but the scope seems to be too broad.
- The program and the scales used need more clarity.
- Given the number of statistical analyses performed, should use more caution in interpreting the findings.
Below are the detailed comments:
- What exactly the program does to prevent suicide is rather vague. Although “networks” is defined clearly, the composition and functions of these networks are still not clear.
- Section 1 should provide some literature review on the variables/predictors measured in the study. Hypotheses pertaining to the study aims should be included as well.
- An ideal design of the study should be a nested model, with participants nested within networks. Collecting the perspectives of program coordinators seems reasonable as well, despite succumbing to several weaknesses. This point is worth some discussion in the Discussion section.
- Section 2.1 does not describe the study design. I believe the study should be a cross-sectional correlational study, instead of a “cross-sectional online survey.”
- Section 2.2 may benefit from more details related to the survey administered, e.g., the number of items for each variable and the psychometric properties of the survey.
- Sections 3.1 to 3.4 should be trimmed down since the statistics are summarized in tables. Also, the interpretations of these findings should use more cautions because no inferential statistical analyses were performed, and there was a lack of comparison (before vs. after, or treatment vs. control).
- Sections 3.5 and 3.6 involve at least 108 statistical analyses, and less than half of the findings were significant. The findings could be merely by chance. The authors are advised to perform a Bonferroni correction and to trim down the number of analyses.
- The arguments in Section 4 should be discussed in relation to existing literature (note that p.11 does not include any references). Section 4.1 should be enriched.
Author Response
Dear Editors and Reviewers,
We greatly appreciate the constructive feedback on our manuscript ‘What works in community-led suicide prevention: Perspectives of Wesley LifeForce network coordinators’.
For your consideration, please now find the revised manuscript attached, alongside a point-by-point response addressing the reviewer feedback below. All resulting changes to the revised manuscript have been highlighted in tracked changes. We believe that the revised manuscript has further gained in clarity and been strengthened considerably.
On behalf of the authors.
Yours sincerely,
Dr Lennart Reifels
Reviewer 2:
This manuscript, titled “What works in community-led suicide prevention: Perspectives of Wesley LifeForce network coordinators,” examined the predictors of the effectiveness of the Wesley Lifeforce Suicide Prevention Networks program. The researchers recruited the coordinators of these networks who provided their perspectives on the factors that determined the outcomes of the program. The results identified several significant predictors, such as strategic planning and additional funding.
The study examined an important research question. However, there are several major issues that need to be addressed.
- The study should be theory-driven, and the literature review should discuss in relation to theories and past literature. The study had clear aims but the scope seems to be too broad.
- The program and the scales used need more clarity.
- Given the number of statistical analyses performed, should use more caution in interpreting the findings.
RESPONSE:
Thank you for the constructive feedback and these helpful suggestions. We note that in view of the nascent state of research on community coalitions in the field suicide prevention, the primary aims of our study were largely exploratory in nature and not formally hypothesis driven. However, further background information on the study links to the broader existing literature and theory on community coalitions, as well as the program and key study variables has now been incorporated in relevant sections of the revised manuscript, as outlined in more detail in our response to specific feedback points below.
Below are the detailed comments:
- What exactly the program does to prevent suicide is rather vague. Although “networks” is defined clearly, the composition and functions of these networks are still not clear.
RESPONSE:
Thank you for sharing this observation. By way of clarification, we note that by contrast to community-based suicide prevention programs with predetermined intervention components, what LifeForce Networks do to prevent suicide is very much determined at a local community-level and therefore does vary across networks and communities. As such, there is no fixed or standard intervention program that could be outlined and that would uniformly apply across the board. This study therefore partly aimed to develop a national profile of key network characteristics, internal network processes and community focused network activities (as outlined in Tables 1 and 2 and Sections 3.1. and 3.2.). In other words, networks provide a structural mechanism to develop and implement locally relevant solutions, rather than providing a fixed program of interventions. To further clarify this aspect, the latter point has now been reflected in the introduction. Further information on the typical composition of networks is available in Section 1.1.
- Section 1 should provide some literature review on the variables/predictors measured in the study. Hypotheses pertaining to the study aims should be included as well.
RESPONSE:
In view of the nascent state of research on community coalitions in the field suicide prevention, the aims of our study were largely exploratory in nature and not formally hypothesis driven. The survey tool was therefore customised to the unique context and activities of suicide prevention networks. Further background information on the scarce existing research in this area, the key variables considered in the broader community coalition literature, and specific variables used in the present study, has now been incorporated in Sections 1 and 2.
- An ideal design of the study should be a nested model, with participants nested within networks. Collecting the perspectives of program coordinators seems reasonable as well, despite succumbing to several weaknesses. This point is worth some discussion in the Discussion section.
RESPONSE:
Thank you for raising this point. The potential for future research to solicit perspectives of multiple respondents within each network has now been reflected in Section 4.1., which reads:
“The survey data are directly informed by the perspectives of network coordinators who are intimately involved in running the network operations, and as such do not provide a direct account of the views of or impacts on community members, service providers or those at risk of suicide. Future research may therefore benefit from soliciting the perspectives of multiple respondents within each network, and from triangulating survey findings with other data sources on network internal and community outcomes [11].”
- Section 2.1 does not describe the study design. I believe the study should be a cross-sectional correlational study, instead of a “cross-sectional online survey.”
RESPONSE:
This point has now been clarified and corrected in Section 2.1.
- Section 2.2 may benefit from more details related to the survey administered, e.g., the number of items for each variable and the psychometric properties of the survey.
RESPONSE:
Section 2.2. has now been updated to incorporate further information on the number of survey items per outcome variable, while information on the psychometric properties (i.e., internal consistency) of composite outcome scales has been included in Section 2.4.
- Sections 3.1 to 3.4 should be trimmed down since the statistics are summarized in tables. Also, the interpretations of these findings should use more cautions because no inferential statistical analyses were performed, and there was a lack of comparison (before vs. after, or treatment vs. control).
RESPONSE:
Sections 3.1. and 3.4. were trimmed down to minimize duplication with statistics reported in tables. More cautious wording was used in interpreting these results in Section 4. Section 4.1. now incorporates additional study limitations of relevance to perceived network outcomes (i.e., lack of before and after assessments and control group comparisons).
- Sections 3.5 and 3.6 involve at least 108 statistical analyses, and less than half of the findings were significant. The findings could be merely by chance. The authors are advised to perform a Bonferroni correction and to trim down the number of analyses.
RESPONSE: Thank you for raising this important point. As mentioned, in view of the dearth of prior research on community coalitions in the field of suicide prevention, our study was largely exploratory in nature and not formally hypothesis driven. As such, we applied a broad lens to identify those network key characteristics, processes and activities that were associated with perceived network member and community outcomes. While we appreciate that the paper contains a large number of analyses, which is not uncommon for exploratory studies, with an alpha of .05 we would generally only expect 5% to be chance findings. Nevertheless, to incorporate this important point in the paper, we have now rerun all inferential analyses, correcting for multiple comparisons, using the Holm method. Sections 2.4., 3.5, 3.6. and Tables 5 & 6 were updated accordingly to reflect a more cautious approach and the adjusted results.
- The arguments in Section 4 should be discussed in relation to existing literature (note that p.11 does not include any references). Section 4.1 should be enriched.
RESPONSE:
Section 4 now includes further references to the existing literature and discussion of study findings, while Section 4.1. has been enriched.
Round 2
Reviewer 1 Report
The authors' replies seem reasonable and lead to more clarifications of the issues. This new version has a number of improvements. I have no comments about it. Please make sure that the entire text is well edited.
Author Response
Dear Editors and Reviewers,
Thank you again for the minor remaining comments on our manuscript. Please now find a revised manuscript attached which addresses these comments as well as a point-by-point response below.
On behalf of the authors.
Yours sincerely,
Dr Lennart Reifels
Reviewer 1
The authors' replies seem reasonable and lead to more clarifications of the issues. This new version has a number of improvements. I have no comments about it. Please make sure that the entire text is well edited.
RESPONSE:
Thank you again for your comments. The entire manuscript text has been checked throughout.
Reviewer 2 Report
The authors are commended for the changes made. However, two issues remain.
- The study should be theory-driven, and the literature review should discuss in relation to theories and past literature. The study had clear aims but the scope seems to be too broad.
- Need a more concrete description of community-led suicide prevention programs and the scales used.
Author Response
Dear Editors and Reviewers,
Thank you again for the minor remaining comments on our manuscript. Please now find a revised manuscript attached which addresses these comments as well as a point-by-point response below.
On behalf of the authors.
Yours sincerely,
Dr Lennart Reifels
Reviewer 2
The authors are commended for the changes made. However, two issues remain.
- The study should be theory-driven, and the literature review should discuss in relation to theories and past literature. The study had clear aims but the scope seems to be too broad.
RESPONSE:
Thank you again for the earlier comments and for raising these remaining points. Research on community coalitions and community networks is not an area that is strongly theory driven and as such there is no overarching theory explaining how they work. In this context, Cunningham et al. (2019) observed that “… despite the growth in research on network effectiveness since the early 1990s, many authors note the lack of widely accepted theories about network effectiveness and its determinants.”
As previously mentioned, in view of the nascent state of research on community coalitions in field of suicide prevention, more specifically, our study was therefore largely exploratory in nature and not formally hypothesis driven. We applied a broad lens to examine those key factors which have previously been investigated in the community coalition literature, whilst adopting customized scales tailored to the unique context of suicide prevention networks. The study design was further underpinned by a program logic for LifeForce networks, which outlined key internal network processes and member outcomes as well as community-focused activities and related community outcomes. To further clarify this circumstance and the fact that our study built and expanded upon prior research and key factors examined in this area, three sections have now been amended as follows.
The final paragraph in Section 1 now reads:
“Among the key factors found to underpin the effective functioning and internal and external outcomes of community and health coalitions are formal characteristics of coalition structure and processes of internal organisation [10,15]. Moreover, reviewing the literature on health networks, Cunningham et al. concluded that “more work is needed to demonstrate effectiveness, and to identify factors related to improved outcomes” [16]. Beyond a prior research emphasis on formal network characteristics underpinning network effectiveness, it is therefore essential to further examine those community-focused suicide prevention activities which are associated with improved community outcomes [9].”
In addition, Section 2.1. now clarifies that:
“The parameters of interest were initially informed by prior research literature on community coalitions and a program logic for LifeForce networks, outlining key internal network processes and anticipated or ‘expected’ member outcomes, community-focused network activities and related community outcomes.”
Moreover, Section 4 now states that:
“The program logic of the LifeForce networks informed, and was subsequently informed by, this study. As such, this research adds to the developing theory of change on suicide prevention networks.”
2. Need a more concrete description of community-led suicide prevention programs and the scales used.
RESPONSE:
The relevant paragraph on community-led LifeForce networks in Section 1.1. has been amended to outline the common portfolio of community-focused network activities. It reads:
“Wesley Mission defines a network as “a union of people and organisations, working together to change the outcome relating to a specific problem.” Networks are further described as being community-led and as being “for the people, by the people” [17]. While the aims and objectives of each network vary, reflecting the unique identity of each community, there is a common thread: a focus on interagency cooperation and raising community awareness. Their portfolio of activities is uniquely tailored to local community contexts and commonly aimed at upstream initiatives that are focused on awareness raising, stigma reduction, fostering help-seeking, training, capacity building and community development.”
In addition, Table 2 outlines the key community-focused network activities examined in this study.
While striving to contain the overall manuscript length, Section 2.2. has been further amended to provide greater clarity around the survey tool and scales used (incorporating cross-references to relevant scale items outlined in tables). It now reads:
“The customised survey included a mix of closed and open-ended questions to capture information on respondent demographics, network characteristics, internal network processes, community-focused activities, as well as perceived outcomes for network members and the community. Respondents were asked whether their network had conducted any of a series of nine internal network processes or nine community-focused activities (see Table 2). Next, they were asked whether they had observed any changes resulting from these network processes and activities for network members and the community. The two main internal network outcomes examined in this study included perceived changes in network member understanding of key suicide prevention issues as well as member confidence and capacity to address such issues. Each of these network member outcomes was assessed through eight or seven survey items, respectively (see Table 3). Perceived external outcomes of network activities for communities, service providers, people at risk of suicide, and overall since the network began, were also examined, by way of four or five survey items, respectively (see Table 4). Respondents indicated on 5-point Likert scales to what extent they agreed with observing changes in these outcome domains (with ratings being 1: strongly disagree, 2: disagree, 3: neutral, 4: agree, 5: strongly agree). A copy of the survey schedule is included in Supplementary Material S1.”